# The Practice Guidelines for Multidose Drug Dispensing Need Revision—An Investigation of Prescription Problems and Interventions

**DOI:** 10.3390/pharmacy9010013

**Published:** 2021-01-06

**Authors:** Anette Vik Josendal, Trine S. Bergmo, Anne Gerd Granas

**Affiliations:** 1Norwegian Centre for E-health Research, University Hospital of North Norway, 9038 Tromsø, Norway; Trine.Bergmo@ehealthresearch.no (T.S.B.); a.g.granas@farmasi.uio.no (A.G.G.); 2Department of Pharmacy, Section for Pharmaceutics and Social Pharmacy, University of Oslo, 0316 Oslo, Norway; 3Department of Pharmacy, Faculty of Health Sciences, UiT The Artic University of Norway, 9037 Tromsø, Norway

**Keywords:** multidose drug dispensing, prescribing errors, pharmacy practice, pharmacist interventions, Norway

## Abstract

Multidose drug dispensing (MDD) is an adherence aid used by one-third of patients receiving home care services in Norway. The system can increase patient safety by reducing dispensing errors and increase adherence, however it has also been criticised for unclear routines and distribution of responsibilities. We investigated prescription problems which pharmacists have detected, and the responsibilities they adopt regarding MDD. For two consecutive weeks, 11 pharmacies used a self-completion form to register prescription problems identified with MDD. Of the 4121 MDD prescriptions, problems were identified on 424 (11%). The most common issues were expired prescriptions (29%), drug shortages (19%), missing prescriber signatures (10%) and unclear/missing medication names or strengths (10%). Compared to ordinary prescriptions, the pharmacist took on additional responsibility for renewing MDD prescriptions. However, because these patients received their medications via the home care service, there was limited patient counselling during dispensing. To increase the efficiency and patient safety of the MDD system, the roles and responsibilities of the pharmacist, GP, and home care nurses in the MDD system should be clearly defined. This seems most urgent for the renewal of prescriptions and patient counselling, where the responsibilities and work practice seem to differ from ordinary prescriptions.

## 1. Introduction

Prescribing errors commonly cause preventable medication errors and adverse drug events in primary care, many of which can lead to patient harm [1,2,3]. Pharmacists increase medication safety by resolving prescription errors and other prescription problems, such as drug shortages, issues with reimbursement, and drug–drug interactions [4,5,6,7,8]. Most studies from primary care show that pharmacists intervene on 0.5–9% of prescriptions dispensed [8,9,10,11], but some show frequencies up to 50% [9,12]. The great variation in these rates is probably due to differences in the definitions of pharmacist’s intervention and in the methods used to register them.

Previous studies of prescription interventions have not included prescriptions for multidose drug dispensing (MDD). MDD, which is used by one third of patients receiving home care services in Norway [13,14], is an adherence aid where the patients’ medications are machine-dispensed in disposable plastic bags, usually for 14 days at a time. MDD is believed to increase medication safety by reducing dispensing errors, reducing discrepancies between medication lists, and increasing adherence [15,16,17,18,19]. However, researchers have raised concerns that the MDD system increases the risk of inappropriate prescribing and medication errors, due to the automation of prescribing procedures and insufficient routines for updating the MDD prescriptions [20,21,22,23].

The same laws apply to MDD and ordinary prescriptions, but there are some practical differences between the two prescription types. In Norway, over 90% of ordinary prescriptions are issued electronically [24], while MDD prescriptions are paper-based and faxed to the pharmacy. In addition, MDD prescriptions are often printouts from a GP’s medical journal and contain a complete list of the patients’ medications, including regular medications, as needed medications, and dietary supplements. This differs from ordinary electronic prescriptions, where a prescription consists of only one item at a time. The only legal difference between the two prescriptions is the timing of the pharmacist’s check [25]. For ordinary prescriptions, the pharmacist checks the prescription at each dispensing, usually every three months for repeat prescriptions, while the MDD prescription is only checked when there are changes in the patient’s medication treatment.

Regardless of prescription type, the pharmacist’s checks ensure that the medication, dosage form, and the dose prescribed is in accordance with the patient’s age, gender and indication written on the prescription. The pharmacist also checks for interactions, contraindications, and other available information about the patient. In addition, the validity of the prescription, including the prescriber’s identity and right to prescribe medications is checked. Lastly, the pharmacist assesses whether the prescription label is clearly written and whether the patient needs any additional information [25,26]. If the prescription is incomplete, unclear or contains other problems, the pharmacist should try to correct the error and intervene. If this is not possible, an emergency refill can be dispensed if the pharmacist deems it necessary to prevent gaps in treatment and patient harm [25].

Most problems with the prescriptions are identified during the pharmacy check, however, the pharmacist’s ability to detect and resolve problems on prescriptions is affected by factors such as patient age, care-setting, types of prescriptions and pharmacy [5,8,9,10,27,28,29,30,31]. Given the differences in format and routines for dispensing MDD and ordinary prescriptions, it is likely that the pharmacist’s intervention rates vary between these two prescription types. Despite the MDD system being criticised for its vague distribution of responsibility and unclear routines across professional borders [17,32,33,34], no previous studies have investigated the pharmacist’s responsibilities in the MDD system.

This study investigates: (1) the prescription problems in which pharmacists intervene; and (2) the responsibilities they adopt while dispensing MDD prescriptions in Norwegian community pharmacies.

## 2. Materials and Methods

### 2.1. The Development and Testing of the Registration Form

This cross-sectional study provides a descriptive analysis of prescription problems and pharmacist interventions on MDD prescriptions. The term “prescription problem” refers to errors, ambiguities, omissions, or other problems with the prescription that are potentially harmful to patients or interfere with the dispensing process. “Pharmacist intervention” refers to any action the pharmacist takes to resolve these prescription problems.

Based on a form used on ordinary prescriptions in previous studies [4,30], we developed a self-completion form for the registration of interventions on MDD prescriptions (See Appendix A). Prescription problems were grouped as either formal or medication-related (see Table 1) or relating to the MDD order (data not presented in this study). Information about the patient, prescription, interventions, the outcomes of the interventions and time spent on correcting errors were also recorded.

A teaching manual including examples of common prescription problems and completed forms was developed. These were piloted on 100 prescriptions in one pharmacy in 2017. The pilot provided important input to the design and content of the form and teaching manual. The pilot interventions are not included in this study. The first three pharmacies in this study provided feedback that led to minor adjustments in the self-completion form, which were used in the eight consecutive pharmacies.

### 2.2. Selection of Pharmacies

At the time of our study, one pharmacy chain dispensed 80% of all MDD prescriptions in Norway. We asked this pharmacy chain for a list of pharmacies with ≥500 MDD patients. This cut-off was set to obtain a reasonable number of interventions per pharmacy. About 12% of MDD patients have changes in their drug regimen between MDD orders [35], therefore 500 MDD patients equalled approximately 60 MDD prescription checks per week. The pharmacy chain provided a list of 35 pharmacies. We purposefully selected 20 pharmacies and invited them by email to participate in our study. We aimed to obtain a variety of geographical representation and workloads in terms of the number of MDD prescriptions dispensed.

### 2.3. Data Collection

The study was conducted from February to October 2018. Each pharmacy collected data for two consecutive weeks. We chose this time frame because most MDD bags last for two weeks, which means that the pharmacy will order a new delivery of MDD for all their patients within this period. To leave time for adjustments to the self-completion form between the periods, no two pharmacies collected data during the same weeks. Once a pharmacy accepted the invitation, a date for participation was allocated and the teaching manual and self-completion form were emailed to the pharmacy. A week before participation, we contacted the pharmacy to check whether the participants had read the teaching manual and we answered any questions they had about the study.

Using the self-completion form, the participants registered all prescription problems and interventions on MDD prescriptions for two consecutive weeks. The forms were sent back to the researchers by post. The pharmacy chain extracted the total number of MDD prescriptions dispensed from each pharmacy from their central dispensing database. We defined the number of prescriptions as the number of pharmacist prescription checks. This corresponds to the number of patients having changes in their medication treatment during the study period.

The data were analysed in Microsoft Excel 2016. No personally identifiable information was recorded, and thus the study did not require approval from the Regional Ethics Committee.

## 3. Results

Of the 20 pharmacies invited to participate, 11 accepted. The main reason for declining was a lack of time or resources. The participating pharmacies were located in 9 of the 11 counties in Norway, and dispensed between 47 and 1813 MDD prescriptions per week, with a median of 109. In total, the pharmacists intervened on 464 prescriptions, an intervention rate of 11.3% of all MDD prescriptions (*n* = 4121). The intervention rate varied from 2.2% to 38.2% between the participating pharmacies.

As shown in Table 1, half of the interventions were related to formal problems with the prescription, about one-third to medication issues, and one-fifth to drug shortages. On average, the medication-related problems took 8.0 min to correct, drug shortages took 4.5 min and formal prescription problems took 3.6 min. Problems related to drug shortages varied from 0 to 55% of the problems detected in each pharmacy.

Missing or unclear information about medication names, strengths, doses, or schedules accounted for more than half of the medication-related prescription problems, in which the majority were corrected or clarified before dispensing. Table 2 shows that a patient’s medication history was the most common reason for the pharmacist to intervene on a prescription, followed by computer-generated warnings and conflicting information about the patient’s medication use from different sources (e.g., prescriptions from other doctors than the GP, discharge notes from hospitals, messages from home care services, etc.).

Table 1 illustrates that over 70% of the formal prescription problems were due to expired prescriptions or missing prescriber signatures. In addition, the prescribers were contacted regarding 42 prescriptions that were about to expire (data not shown). For the majority of formal prescription problems, the pharmacist did not obtain a valid prescription before sending the MDD order and, therefore, dispensed void prescriptions.

In total, 88% of the problems were resolved by a pharmacist and the remaining by pharmacy technicians. For 55% of the prescription problems, the pharmacist contacted the prescriber to resolve the problem; for 17%, the home care services were contacted; and for another 17%, the pharmacist used their professional judgment to resolve the problem.

Based on the problems detected and actions taken by the pharmacist, we have identified five different responsibilities the pharmacist adopts while dispensing MDD prescriptions: checking prescriptions for clinical appropriateness, verifying the validity of prescriptions, renewing prescriptions, patient counselling, and dispensing emergency refills.

## 4. Discussion

### 4.1. Pharmacist Interventions

One in nine of all MDD prescriptions needs an intervention or clarification by the pharmacist before dispensing. The most common prescription problems were expired prescriptions (27% of all problems), drug shortages (19%), and missing signatures (13%). For most of the medication-related problems, the pharmacist clarified or corrected the problem before dispensing. For the majority of formal prescription problems, the pharmacist reported dispensing MDD despite the prescription being invalid. Five percent of the prescription problems resulted in one or more medications not being dispensed.

To our knowledge, our study is the first to specifically investigate prescription problems on MDD prescriptions. A Danish study reported errors on MDD prescriptions as a part of a larger study and found an intervention rate of 0.85% [11]. This is considerably lower than our findings, but this is expected due to the differences in definitions of an MDD prescription. We have used the number of pharmacist checks (i.e., whenever there are changes on a prescription), while the Danish study used the number of MDD orders (i.e., one prescription every two weeks for each patient).

A prescription intervention rate of 11.3% is high compared to studies in community pharmacies, which usually show interventions on 0.5–9.0% of prescriptions [8,9,10,11]. For MDD prescriptions, the pharmacist usually has more information about the patients, including a complete medication list and the indication for use. This could contribute to a higher intervention rate [5,31]. Some evidence also suggests that intervention rates are higher on new prescriptions than repeat prescriptions [8,30,36]. We have used the number of pharmacist checks (i.e., whenever there are changes in the drug treatment) as the number of prescriptions dispensed, which might also explain our high intervention rate. On the other hand, the pharmacist has less contact with the patients when supplying MDD prescriptions, which could contribute to a lower intervention rate [4,5,27,30,37].

### 4.2. The Pharmacist’s Responsibilities

#### 4.2.1. Checking Prescriptions for Clinical Appropriateness

The medication-related problems in our study constituted 29% of all problems; 3.3% of the total MDD prescriptions. In comparison, clinically relevant interventions are done on fewer than 1% of ordinary prescriptions [5,6,7,9,11,12,30,31]. “Medication-related interventions” is a broader definition than “clinically relevant interventions”, therefore it is expected that our rate is higher than what is shown in these studies. The types of problems detected are similar to other studies, where missing or unclear information about the medication name, strength, dose, or dosing schedule are among the most common problems [4,5,7,11,28,30].

In our study, the pharmacist identified certain problems which would be difficult to detect on ordinary prescriptions. When checking new MDD prescriptions, the pharmacist compares this with the previous MDD medication list. This enables the pharmacist to identify problems such as a sudden stop in medication treatment. In our study, 15% of the medication-related problems were identified using the patients’ medication histories in this way. We also discovered that 13% of the medication-related problems were revealed because the pharmacist had access to different prescriptions with inconsistent information about the patient’s current treatment. On ordinary prescriptions, the pharmacist rarely has access to other information about the patient, apart from the electronic prescriptions they are dispensing.

#### 4.2.2. Verifying the Validity of Prescriptions

The majority of prescription problems detected in our study were related to formal errors, in line with other studies regarding paper-based prescriptions [4,7,10,28,30]. Patient and prescriber data are fields that must be completed to enter prescriptions into the dispensing program, but this is not the case for a missing signature. The frequent detection of this problem indicates that the pharmacists actively check the validity of the MDD prescriptions before dispensing, and do not just detect the problems that physically hinder the dispensing.

#### 4.2.3. Renewing Prescriptions

Most prescriptions for regular medications are valid for one year, including MDD prescriptions. Normally, the patient contacts the GP to renew prescriptions when they are about to expire, but for MDD patients the home care service has this responsibility. However, as our study shows, the pharmacist partly takes on this responsibility. Informing that a prescription was expired (*n* = 125) or about to expire (*n* = 43) was the single most common cause of the pharmacist contacting the prescriber. It seems to have become a common practice for many pharmacies to contact the GPs directly to renew MDD prescriptions to prevent unintentional gaps in drug treatment [17].

#### 4.2.4. Counselling Patients on the Use of Prescription Medications

The pharmacists should assess whether the prescription label is clearly written and whether the patient needs any additional information about their medications [26]. The MDD bags contain more information than medication labels, therefore this might reduce the need for clarifying complex medication regimens, such as informing that medications should be taken some hours apart or at specific times of the day.

However, assessing the patient’s need for information is more difficult in the MDD system than for ordinary prescriptions, because the pharmacist has no direct contact with the patient. Any medication counselling has to be conducted via the home care services. When dispensing ordinary prescriptions, contact with the patient or caregiver can resolve up to half of the prescription-related problems [4,5,30,37]. Contact with the home care nurses resolved only 17% of the problems in our study, while the majority were resolved by contacting the GP. Our study indicates that the pharmacists only take limited responsibility for patient counselling when supplying MDD prescriptions. Reduced patient counselling about prescription medications upon dispensing might explain why patients using MDD have less knowledge about their medications than patients with ordinary prescriptions [16,18].

#### 4.2.5. Dispensing Emergency Refills

Although the pharmacist frequently intervened on formal prescription problems, only 40% of these problems were corrected before dispensing, while 54% were dispensed despite the prescription being invalid. Previous studies also show that for formal errors, the majority of prescriptions were dispensed as prescribed [4,30]. However, the problems detected in these studies were mostly reimbursement issues and missing patient information, which were not errors that made the prescriptions invalid.

We have not investigated why the rate of dispensing invalid prescriptions is so high in our study; however, there are some important differences between the MDD system and ordinary prescriptions that might explain this finding. Stopping an MDD prescription involves stopping all medications distributed in MDD, not just a single medication. Secondly, MDD patients usually only have a few days of medication supply left when the new MDD order is placed. The MDD bags have to be produced and shipped before the patient receives them, therefore this leaves the pharmacist with less time to correct errors and omissions on the prescriptions before the patient runs out of the medications. These factors might raise the pharmacist’s threshold of stopping the dispensing.

According to legislation, a pharmacist can dispense an emergency refill only once per prescription, and only in amounts necessary until the prescriber can be reached and the error corrected [25]. If the emergency refill is only performed once for 14 days, this can prevent patient harm because it prevents gaps in the patient’s regular medications. However, if this practice is continued for longer periods, it might ultimately reduce patient safety due to the patient not receiving regular medication reviews. If dispensing emergency refills is a common practice, this might also explain why MDD users seem to have fewer changes in their medication treatment than patients with ordinary prescriptions [20].

### 4.3. Implications and Suggestions for Improvement

This study shows that pharmacists play an important role in detecting and resolving problems in MDD prescriptions. The pharmacist’s responsibility and practice for checking clinical appropriateness and the validity of prescriptions seems to be similar for ordinary prescriptions and MDD prescriptions. However, one can speculate whether the pharmacist should have detected even more medication-related problems for these patients. MDD patients, in general, use many medications, and potentially inappropriate medications are common [20,38]. Pharmacists have access to the complete medication list for these patients; this puts them in a position to review the medication treatment as a whole.

The pharmacist is responsible for counselling patients on the use of prescription medications; however, this seems to happen only to a limited degree for MDD patients. There is no direct contact between the patient and the pharmacist upon dispensing, therefore increasing the use of E-health technology, such as video consultations between the patient and the pharmacist, could be considered. In the MDD system, the home care services are also responsible for patient counselling [39]. To ensure that the patients receives the information they need, the roles and responsibilities for patient counselling between these professional groups should be more clearly defined.

We see that pharmacists frequently dispense emergency refills for MDD patients, mostly due to prescriptions being expired. Additionally, pharmacists seem to have taken on the additional responsibility of contacting GPs to renew prescriptions, a responsibility which they do not take with ordinary prescriptions. This indicates that there is a lack of clear guidelines or enforcement of such guidelines when it comes to the renewal of prescriptions in the MDD system. This highlights the need for clarifying the responsibility of the GP, home care nurses and the pharmacist in this setting.

### 4.4. Strengths and Limitations of the Study

A strength in our study is that our registration form was based on those in previous studies, and was piloted before the study. However, a self-completed form means that problems might be underreported. Our selection of pharmacies is not representative for MDD pharmacies in Norway, which means that there is uncertainty in the frequency of different types of prescription problems. However, we feel that the purposeful selection helped us capture the different types of problems which pharmacists identified. We recruited pharmacies from all parts of the country, which increased the generalisability, although there might have been a selection bias because many pharmacies declined to participate. We were investigating routines for managing prescription problems, therefore only having pharmacies from one of three chains is a limitation. However, we found a great variation in both the number of problems detected and how they were solved, which indicates that there is still variation in routines within the pharmacy chain. We also lack the information of any clinical impact of the pharmacists’ interventions.

## 5. Conclusions

Community pharmacists contribute to safe and effective drug use by clarifying problems in one of every nine MDD prescriptions. One-third of the problems were related to medications, half were formal errors with the prescriptions and the remaining problems were related to drug shortages. As for ordinary prescriptions, the pharmacists check prescriptions for clinical appropriateness, verify the validity of prescriptions, and dispense emergency refills when deemed necessary. However, the pharmacists seem to take on additional responsibilities for renewing prescriptions and counsel patients less than for patients without MDD prescriptions. The responsibilities of the pharmacist differ between ordinary prescriptions and MDD prescriptions, therefore there is a need for specific practice guidelines for dispensing MDD prescriptions. Clearly defining the roles and responsibilities of the pharmacist, GPs, and home care nurses, especially regarding the renewal of prescriptions and patient counselling, has the potential to increase efficiency and patient safety of the MDD system.

## Figures and Tables

**Table 1 pharmacy-09-00013-t001:** Prescription problems detected (*n* = 464), and the result of pharmacist interventions on multidose drug dispensing (MDD) prescriptions.

	Result of Intervention						
	*n* (Percentage of Problem Type)						
Problem with Prescription	Changed or Clarified Prescription	Dispensed as Prescribed	Prescription Not Dispensed	Other	Sum		Percentage of Total
**Medication-related problems**	**104 (77%)**	**16 (12%)**	**2 (1%)**	**13 (10%)**	**135**	**(100%)**	**29%**
Medication name or strength	42	1	2	0	45	(33%)	10%
Dose or schedule	37	2	0	2	41	(30%)	9%
Drug–drug interaction	9	9	0	2	20	(15%)	4%
Administration formula	7	0	0	0	7	(5%)	2%
Treatment duration	6	1	0	0	7	(5%)	2%
Other	3	3	0	9	15	(11%)	3%
**Formal prescription problems**	**93 (39%)**	**131 (54%)**	**8 (3%)**	**9 (4%)**	**241**	**(100%)**	**52%**
Prescription date expired	8	125	0	0	133	(55%)	29%
Missing signature from the prescriber	39	1	7	1	48	(20%)	10%
Reimbursement	22	1	0	1	24	(10%)	5%
Missing prescriber data	9	0	0	2	11	(5%)	2%
Missing patient data	8	0	0	0	8	(3%)	2%
Other	7	4	1	5	17	(7%)	4%
**Drug shortages**	**52 (59%)**	**21 (24%)**	**13 (15%)**	**2 (2%)**	**88**	**(100%)**	**19%**
**TOTAL**	**249 (54%)**	**168 (36%)**	**23 (5%)**	**24 (5%)**	**464**	**(100%)**	**100%**

**Table 2 pharmacy-09-00013-t002:** The five most frequent reasons for pharmacist intervention on medication-related problems on MDD prescriptions.

Reason for Intervention	Examples of Prescription Problems	*n* (%)
Patient medication history	The new prescription stated candesartan 32 mg, while the previous prescription was 8 mg. The pharmacist reacted to the sudden change in dose. The patient had used methylphenidate 54 + 36 mg daily, but the medication was missing on the new prescription. The pharmacist wondered if the stop was intentional.	21 (16%)
Computer-generated warnings (drug–drug interactions)	Iron tablets and calcium were prescribed to the same patient at the same time of day. Calcium reduces the absorption of iron, and these should be taken 2–3 h apart from each other. A patient started escitalopram while he was already using dabigatran. This increases the chances of bleeding.	20 (15%)
Multiple sources with conflicting information about the prescribed medicines	On the paper-based MDD prescription, ticagrelor was prescribed as 90 mg, one tablet daily. On an ordinary electronic prescription, the dosing schedule was 90 mg, two times daily. The pharmacist wondered which dose the patient should have. Discharge notes from the hospital stated a temporary reduction in the dose of apixaban. On the MDD prescription from the GP, the treatment of apixaban was stopped. It was unclear which of the prescriptions were the newest/correct.	18 (13%)
Incomplete prescriptions (due to handwriting)	A patient was prescribed valsartan, one tablet daily. No strength was given. Prescribed “iron tablets”. The strength and dose were lacking.	17 (13%)
Other inconsistencies	A patient was prescribed a high dose of prednisolone in MDD without a stop date. The pharmacist called to check if there should be a tapering schedule. A prescription contained two different dosing schedules for the same medication. Unclear which one was correct.	17 (13%)

## Data Availability

The data underlying this article will be shared on reasonable request to the corresponding author.

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
