# Peer review of "The Practice Guidelines for Multidose Drug Dispensing Need Revision—An Investigation of Prescription Problems and Interventions"

_pharmacy, 2021, doi:10.3390/pharmacy9010013_

Round 1
Reviewer 1 Report
- The term ‘prescription problems’ should be defined earlier in the Introduction or later in the Methods section.
- The authors need to highlight what modifications were made to the forms used in previous studies and why these were useful for the current study?
- I would suggest for the authors to include (as an appendix) a copy of the self-completing form used to collect data to increase the transparency and understanding of the type of data collected.
- A pharmacy with a higher number of MDD prescriptions would identify more errors. How did the authors account for this variability?
- How was selection bias avoided when identifying pharmacies to participate in the study?
- What was included in a pharmacist ‘intervention’? Please add this definition to the Methods section.
- It would be interesting to find out if there was a designated pharmacist to provide clinical interventions in pharmacies that had a higher load of MDD prescriptions? How did the number of interventions by this pharmacist differ from other pharmacies that did not have a designated pharmacist?
Author Response
Reviewer’s comment: The term ‘prescription problems’ should be defined earlier in the Introduction or later in the Methods section.
What was included in a pharmacist ‘intervention’? Please add this definition to the Methods section.
Reply
We have included both definitions at the beginning of the methods section.
Reviewer’s comment: The authors need to highlight what modifications were made to the forms used in previous studies and why these were useful for the current study?
I would suggest for the authors to include (as an appendix) a copy of the self-completing form used to collect data to increase the transparency and understanding of the type of data collected.
Reply
The final form we ended up using in this study was quite different from the original form used in the previous studies we have cited. I have therefore rewritten this part to be: “We developed a self-completing form for registration of interventions on MDD prescriptions, based on a form used in previous studies” rather than stating that we modified the form.
We have also attached the complete form as an appendix.
Reviewer’s comment: A pharmacy with a higher number of MDD prescriptions would identify more errors. How did the authors account for this variability?
Reply
Since this is a descriptive study without modelling or comparison of pharmacies, we have not seen the need to account for this variability.
We chose to collect data over 2 weeks from all pharmacies because most MDD bags last two weeks. This means that the pharmacy will order MDD for all their patients within this period. Having some pharmacies collect for longer periods would mean that we risk repeat registering of the same problems. The reason for our choice of 2 weeks of data-collections has also been added to the “data collection” section of the manuscript.
Reviewer’s comment:
How was selection bias avoided when identifying pharmacies to participate in the study?
Reply
Selection bias is a limitation to our study. First, we only contacted the one pharmacy chain (who produced 80 % of the MDDs in Norway at the time of the study), as mentioned in the limitations this is unfortunate because different chains might have different routines for handling prescriptions and prescription problems. The proportions of different types of problems might therefore not be representative. In addition, many pharmacies declined our invitation.
In our initial selection, we purposefully chose pharmacies of different sizes and locations. We consider it likely that we have identified the different routines this pharmacy chain has for dispensing and correcting MDD prescriptions.
We have added this to our strength and limitation section.
Reviewer’s comment:
It would be interesting to find out if there was a designated pharmacist to provide clinical interventions in pharmacies that had a higher load of MDD prescriptions? How did the number of interventions by this pharmacist differ from other pharmacies that did not have a designated pharmacist?
Reply
There were no designated pharmacist to provide clinical interventions on prescriptions in any of the participating pharmacies.
Reviewer 2 Report
The topic is important. However, flow in the paper is missing.
Please consider to reduce the document with relevant information only - there are redundancies throughout the paper. You may consider the following format: Introduction/current status/problem (comparing with non multi-dose rx), findings, discussion (include specific solutions to prevent patient harm).
The article needs to be re-written, concise and to the point. For example, - Introduction: current dual process , problem with the multidose inititiative. Then objective of their analysis - Method - Result - Discussion a. findings confirming objective b. Specific solutions/respoinsibility/implementation (may use bullet points)
Author Response
The topic is important. However, flow in the paper is missing.
Please consider to reduce the document with relevant information only - there are redundancies throughout the paper. You may consider the following format: Introduction/current status/problem (comparing with non multi-dose rx), findings, discussion (include specific solutions to prevent patient harm).
The article needs to be re-written, concise and to the point. For example, - Introduction: current dual process , problem with the multidose inititiative. Then objective of their analysis - Method - Result - Discussion a. findings confirming objective b. Specific solutions/responsibility/implementation (may use bullet points)
Reply
We have improved the flow between paragraphs, especially in the introduction. We have also clarified that we have a dual aim of both documenting prescription interventions but also to describe the pharmacist’s responsibilities during the dispensing process, which should make our reasoning in both introduction and discussion clearer. We have shortened the manuscript with about 5 % and written it more to the point.
If the manuscript needs more clarification and restructuring, please let us now.
Reviewer 3 Report
Thank you for your submission. It was very interesting. I have a few comments:
Methods: selection of pharmacies: could you reach out to independent pharmacies for this project? Not sure if they are not allowed to process MDD prescriptions. If not. may be explain that independent pharmacies are restricted from MDD prescriptions.
Methods are sound and explained well.
Table 1: Is there a way to make it easier to read? Maybe an indent for each issue under the title. ie
Medication related problems
Medication name or strength
Table 2 - all generic names of medications should be in lower case not with an upper case letter
Thank you for including your strengths and limitations.
References: Should be in AMA format- The journal name should not be written out fully rather use the correct journal abbreviation. Missing DOI numbers for some reference.
The following references are missing the website addresses: #17, 19,24, 26, 39
Correct spacing for reference #17
This was an interesting project with many good suggestions for clarity of policies.
Author Response
Reviewer’s comment: Methods: selection of pharmacies: could you reach out to independent pharmacies for this project? Not sure if they are not allowed to process MDD prescriptions. If not. may be explain that independent pharmacies are restricted from MDD prescriptions.
Reply
There are only a few sites in Norway that manufacture MDD (2 or 3), and these are owned by the pharmacy chains. When patients in home care services get MDD that is because the municipality has signed a contract with a manufacturer. Legally, independent pharmacies can process MDD prescriptions, but it would require them buying the service from a competing chain. It is thus not economical. There are a few examples of hospital pharmacies delivering MDD to home care, but this is normally not possible when there are big contracts with many patients, as their production site is too small.
I have not been able to explain this in short in the manuscript. I will keep this in mind for later when I write about the MDD system as a whole.
Reviewer’s comment:
Table 1: Is there a way to make it easier to read? Maybe an indent for each issue under the title.
Reply
We have increased the indents for the subcategories in the table.
Reviewer’s comment:
Table 2 - all generic names of medications should be in lower case not with an upper case letter
Reply
Thank you, we have corrected this now.
Reviewer’s comment
References: Should be in AMA format- The journal name should not be written out fully rather use the correct journal abbreviation. Missing DOI numbers for some reference.
Reply
I have added DOI to those who was lacking this, and abbreviated the journal names.
Reviewer’s comment
The following references are missing the website addresses: #17, 19,24, 26, 39
Reply
For the references that are webpages I have added the name of the webpage, for those who are reports, I’ve added the name of the publisher.
Reviewer’s comment
Correct spacing for reference #17
Reply
Changed the link to #17, so that spacing is not an issue
Round 2
Reviewer 2 Report
The paper provides valuable information on current process of multi-dose drug dispensing and associated medication safety problems. The study is well done.